# TASKWEB: Selecting Better Source Tasks for Multi-task NLP

**Joongwon Kim[†], Akari Asai[†], Gabriel Ilharco[†], Hannaneh Hajishirzi[†♡]**
[†]University of Washington   [♡]Allen Institute for AI
{jwonkim,akari,gamaga,hannaneh}@cs.washington.edu

## Abstract

Recent work in NLP has shown promising results in training models on large amounts of tasks to achieve better generalization. However, it is not well-understood how tasks are related, and how helpful training tasks can be chosen for a new task. In this work, we investigate whether knowing task relationships via pairwise task transfer improves choosing one or more source tasks that help to learn a new target task. We provide TASKWEB, a large-scale benchmark of pairwise task transfers for 22 NLP tasks using three different model types, sizes, and adaptation methods, spanning about 25,000 experiments. Then, we design a new method TASKSHOP based on our analysis of TASKWEB. TASKSHOP uses TASKWEB to estimate the benefit of using a source task for learning a new target task, and to choose a subset of helpful training tasks for multi-task training. Our method improves overall rankings and top-$k$ precision of source tasks by 10% and 38%, respectively. We also use TASKSHOP to build much smaller multi-task training sets that improve zero-shot performances across 11 different target tasks by at least 4.3%. [1]

## 1 Introduction

Recent studies have revealed that large language models are able to generalize to unseen tasks when jointly trained on many different tasks, with their performance scaling to the size and diversity of the training data (Sanh et al., 2022; Wang et al., 2022b; Wei et al., 2022a; Chung et al., 2022; Longpre et al., 2023). As more and more tasks are added to build general-purpose models, it has been noted that knowing inter-task relationships may be helpful but that it remains unclear how to select helpful tasks for multi-task learning (Ye et al., 2021; Min et al., 2022; Asai et al., 2022; Chan et al., 2022).

In this work, we investigate whether quantifying the relationship between different NLP tasks via

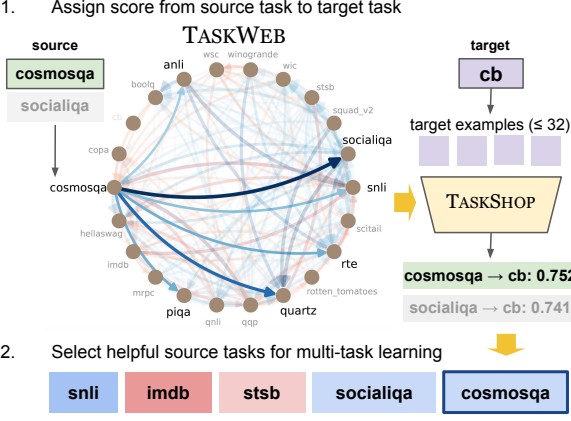

Figure 1: We use pairwise transfer scores in TASKWEB to score (source, target) pairs where the source task is in TASKWEB and the target task is unseen (i.e., access to only a few examples). Then, we select helpful tasks and perform multi-task learning for the target task.

pairwise task transfer helps *task selection*, which we define as choosing one or more source tasks that better initialize a model for an unseen target task as shown in Figure 1. We begin from a pairwise setup as it is often used to quantify task relationships (Zamir et al., 2019; Vu et al., 2020) and is more tractable than larger combinations of tasks.

First, we construct TASKWEB, a large-scale benchmark for pairwise task transfers across different model architectures (encoder-only, decoder-only, encoder-decoder), parameter count (60M to 770M) and adaptation methods including finetuning, Adapter-tuning (Houlsby et al., 2019) and Bit-Fit (Zaken et al., 2022), resulting in 25,000 transfers. From our results, we discover a *transitive* property where having strong, positive transfers A → B and B → C for tasks A, B and C makes it more likely that A → C is also a positive transfer.

Then, we introduce a new method TASKSHOP that predicts the transferability from a source task to a target task associated with only a few examples. TASKSHOP builds upon the transitive behavior to

---

[1]Our code is available at https://github.com/danieljkim0118/TaskWeb.

construct different paths with "pivot" tasks between the source and target tasks. It combines TASKWEB scores between the source and pivot and textual similarity scores between the pivot and target to estimate (source→target) transfers.

We evaluate our methods in both single-task and multi-task settings. First, we show that TASKSHOP assigns better transferability scores both in terms of the overall ranking and identifying top helpful tasks. Then, we demonstrate that models trained on small multi-task sets built with TASKSHOP outperform models trained on larger sets of tasks. We perform additional analyses and discover that there is a tradeoff for building multitask sets of varying sizes with TASKSHOP, and that the proportion of helpful tasks in the training set affects performance.

To summarize, our contributions are as follows:

1. We build and analyze TASKWEB, a benchmark of pairwise transfer experiments across various tasks, models and adaptation methods.

2. We define task selection for single-task and multi-task setups and propose TASKSHOP which uses pairwise transfer scores to predict transfer to an unseen target task.

3. We use TASKSHOP and TASKWEB to choose helpful source tasks and build small multi-task training sets that result in better zero-shot performance for unseen targets.

## 2 Background and Overview

We use pairwise task transfer to quantify task similarities, select better source tasks for unseen tasks and improve performance via multi-task finetuning.

### 2.1 Overview

Figure 2 depicts how we use task relationships to select better source tasks. We first quantify task relations with pairwise task transfer, which is a process of sequentially learning one task—the *source task*—and then another task—the *target task*. We use this to build TASKWEB, a collection of 22 diverse, high-resource tasks in NLP and their pairwise task transfer scores across seven different training setups (Sections 3.1, 3.2). From our analysis, we find that pairwise task transfer indicates *transitive* behavior between positive transfers (Section 3.3).

We then explore task selection, where for a target task $t$ with $n$ examples and a set of source tasks $S$, we select a helpful task $s \in S$ for $t$. Here, we assume that the target task is *unseen*, that is, with

access only to a small number of examples from $t$ ($n \leq 32$). We propose a new task selection method TASKSHOP that builds upon the transitive behavior to select the best source task to transfer to an unseen target task, even without pairwise transfer scores for the target (Section 4.1). We evaluate the overall task rankings and the precision of top-$k$ helpful tasks returned by TASKSHOP (Section 5.1).

Moreover, we extend task selection to a multi-task setup. By selecting tasks $k > 1$ times, we obtain a set of $k$ source tasks as a multi-task training set (Section 4.2). We train models on these multi-task sets and perform evaluations and analyses on 11 different target tasks (Sections 5.2, 5.3).

### 2.2 Related Work

**Pairwise Task Transfer.** Pairwise task transfer, also known as intermediate task transfer, is used to quantify relationships between different tasks in computer vision (Zamir et al., 2019; Achille et al., 2019) and NLP (Vu et al., 2020; Poth et al., 2021). It is also used in NLP to study factors impacting task transfer (Pruksachatkun et al., 2020; Albalak et al., 2022) and identify helpful source tasks for parameter-efficient methods (Vu et al., 2022; Su et al., 2022; Asai et al., 2022). Building upon previous work, we address more diverse tasks, models, and adaptation methods.

**Task Selection.** Task selection is used in many studies to better initialize models for learning new tasks. Some methods assume access to the entire training set and model (Vu et al., 2020; Poth et al., 2021; Vu et al., 2022; Su et al., 2022), while other methods only access a small portion of the training data (Jang et al., 2023; Paranjape et al., 2023). We build upon the second case in this work.

**Multi-task Fine-tuning.** Multi-task fine-tuning is used to train models that generalize across many tasks (Khashabi et al., 2020; Mishra et al., 2022; Sanh et al., 2022). While studies report that adding more tasks generally improve performance, (Aghajanyan et al., 2021; Wei et al., 2022a; Wang et al., 2022b), others report that using a subset of tasks provide better performance (Padmakumar et al., 2022; Chan et al., 2022) but that it is not clear how to identify such subset (Aribandi et al., 2022). Previous work retrieves the top-$k$ relevant source examples based on the target examples (Lin et al., 2022; Ivison et al., 2022). In this work, we take a simpler approach and select helpful *tasks* based on target examples to build multi-task training sets.

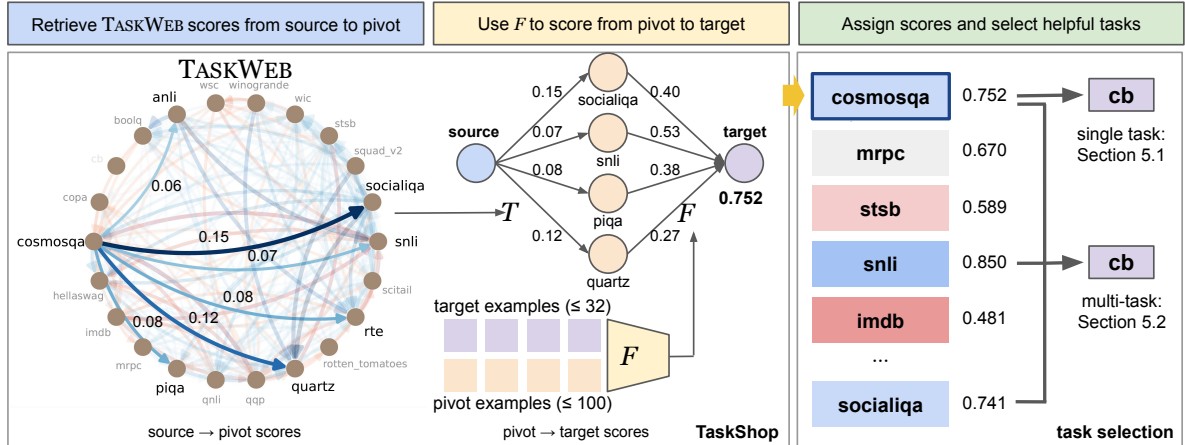

Figure 2: Overview of single and multi-task selection using TASKSHOP and TASKWEB. Section 3 describes the pairwise task transfer involved in TASKWEB as well as its analysis. Section 4 details TASKSHOP and describes task selection in single task and multi-task setups. Section 5 presents our experiments as well as additional analyses.

## 3 TASKWEB: A Benchmark for Pairwise Task Transfer

Previous studies in pairwise task transfer tend to focus on specific models, adaptation methods or task domains (Vu et al., 2020; Poth et al., 2021; Albalak et al., 2022). We introduce TASKWEB, which consists of pairwise task transfer experiments that span a wide variety of tasks, models, and adaptation methods. TASKWEB can be used as a benchmark to evaluate task transferability, and as a repository for selecting helpful source tasks (Section 4).

### 3.1 Focus and Experimental Setup

**Tasks.** To build TASKWEB, we choose a set of 22 representative tasks in NLP that span diverse categories and require various forms of knowledge, as shown in Table 1. We perform a total of about 25,000 transfers between all pairs of tasks.[2]

**Training Procedure.** We finetune a pre-trained language model on the full dataset associated with a source task $s$, and further finetune the model on a set of 1,000 random examples of the target task $t$.[3] Then, we compare the performance gain from initializing the model on $s$ to finetuning the model on the same subset of $t$ without starting from $s$. We repeat this process over eight random seeds to reduce variability (Dodge et al., 2020).

**Models.** We study the impacts of three different model architectures on task transfer—T5 (encoder-

| Category | Tasks |
|---|---|
| NLI/Entailment | ANLI, CB, QNLI, RTE, SciTail, SNLI |
| Paraphrase | MRPC, QQP, STSB |
| Sentiment | IMDB, Rotten Tomatoes |
| Commonsense | COPA, CosmosQA, HellaSwag, PIQA, Quartz, SocialIQA, Winogrande |
| Semantics | WiC, WSC |
| QA | BoolQ, SQuAD2.0 |

Table 1: All tasks used in our pairwise transfer experiments, grouped by high-level task categories. Citations for all datasets are provided in Table 8 in the appendix.

decoder; Raffel et al. 2020), GPT-2 (decoder-only; Radford et al. 2019) and RoBERTa (encoder-only; Liu et al. 2019). We use the LM-adapted versions[4] (Lester et al., 2021) of T5-small/base/large, as well as GPT-2 medium and RoBERTa-base.

**Adaptation Settings.** We investigate pairwise task transfer with three widely-adopted adaptation methods—full fine-tuning, Adapter-tuning (Houlsby et al., 2019) and BitFit (Zaken et al., 2022)—while fixing T5-base as the base model.

**Metrics for Task Transferability.** We follow Vu et al. (2020) and use the average percentage change to measure task transfer. Also, we measure the proportion of models with positive transfer across all random seeds. We combine both metrics to account for both the magnitude and consistency of transfers across all random seeds. The formal definition is provided in Section A.1 in the appendix.

---

[2]We use SQuAD2.0 as only a source task due to difficulties associated with running SQuAD evaluation for all transfers.

[3]This number was chosen for the model to not overfit to $t$, but also learn enough from $t$ to provide a measure of how it would perform on the task, in line with previous studies.

[4]The original T5 checkpoints have been trained on datasets that overlap with ours. We aim to separate the effects of multi-task supervised pretraining in our pairwise transfer analysis.

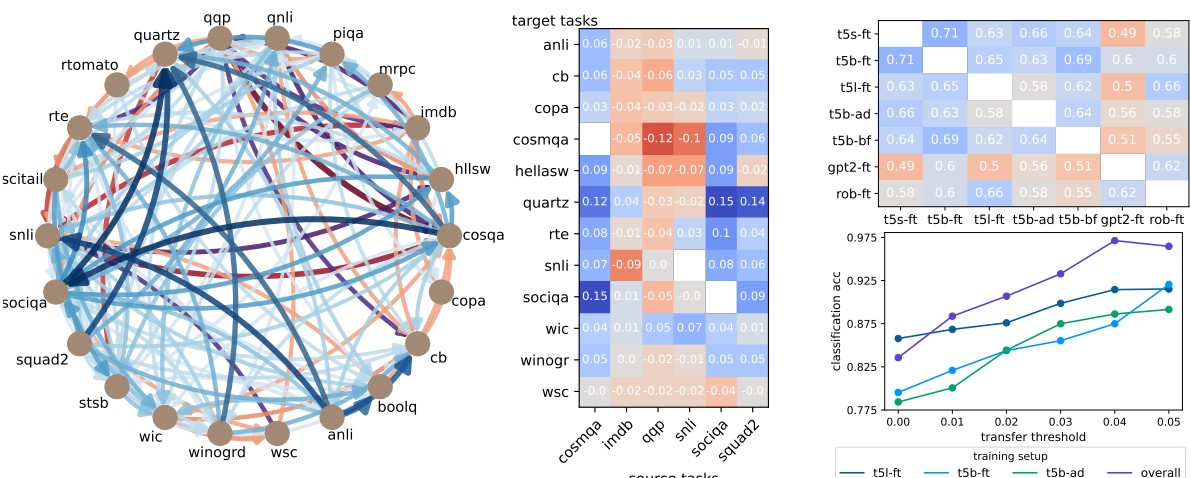

Figure 3: (**Left**) visualization of TASKWEB, our collection of pairwise transfer between 22 different NLP tasks, averaged over seven training setups. Positive transfers are blue and negative transfers are red. All transfers point from the source to the target. (**Center**) transfer scores between a subset of source tasks (three more helpful/three less helpful) and a subset of target tasks. The full set of scores is given in Figure 5 in the appendix. (**Top-right**) similarities between pairwise transfer results in our experiment of 22 tasks obtained for seven different training setups. (**Bottom-right**) probability of identifying positive source → target transfers as the minimum threshold for (source → pivot, pivot → target) transfers is increased. Results with all setups are in Figure 14 in the appendix. t5s/b/l: T5-small/base/large, ft: finetuning, ad: adapter-tuning, bf: BitFit, gpt2: GPT-2 medium, rob: RoBERTa-base.

## 3.2 Observations from TASKWEB

**Results.** Figure 3 visualizes TASKWEB—the left shows all transfers, and the center gives examples of pairwise transfer scores. All scores are averaged over seven training configurations. Refer to Figures 5 to 12 in the appendix for the full results.

We note that positive transfers (blue) occur between intuitively similar tasks such as CosmosQA to SocialIQA (+0.15), both of which are multiple-choice commonsense questions. In contrast, negative transfers (red) occur for tasks that seem to require unrelated skills, such as from QQP to CosmosQA (-0.12). Surprisingly, positive transfers exist between tasks that do not seem similar, such as a positive transfer from SocialIQA to RTE (+0.10).

**Effects of Training Setup.** We investigate how the training setup affects pairwise task transfer. To this end, we build matrices of pairwise transfer scores for each training setup as shown in Figure 5 and compute their normalized dot products.

Refer to the top-right subfigure of Figure 3. We observe more similar pairwise transfers when 1) the same adaptation method is applied to models of the same class but different sizes, or 2) different adaptation methods are applied to the same model. For example, T5-base finetune exhibits more similar transfer with T5-small/large finetune or T5-base adapter/BitFit than GPT-2 or RoBERTa finetune.

## 3.3 Analysis of Mathematical Properties

Computing pairwise transfer scores can become costly as more tasks are added. Would it be possible to predict transferability beforehand using existing scores? We formulate pairwise task transfer as a mathematical relationship and investigate two properties—*commutativity* and *transitivity*.

We define *commutativity* in our setup as whether A → B being a positive/negative transfer implies that B → A is also a positive/negative transfer. If A → B is known, the commutativity would help us predict B → A before performing the transfer.

Meanwhile, we define *transitivity* in our setup as whether knowing the transfer scores of A → B and B → C allows us to infer about A → C. This property would also provide us more flexibility to predict pairwise transfer in advance.

**Commutativity often does not hold.** Based on the pairwise transfer scores shown in Figure 3 (center), we compute the proportion of transfer pairs that exhibit commutativity. Of the 210 unique transfer pairs in our setup, we find that 97 exhibit commutativity. The results are visualized in Figure 13 in the appendix. We uniquely observe from our experiments that pairwise transfer does not display strong signs of commutativity. One possible reason is that while knowledge acquired from task A may be helpful for task $B$, the reverse may not be true.

**Transitivity holds for positive transfers.** We perform a small experiment where we predict transfer A → B as positive if both A → B and B → C score above a threshold. Here, we call A the source task, C the target task, and B the "pivot" task.

Refer to the bottom-right subfigure of Figure 3. We observe that as stricter criteria is imposed for source → pivot and pivot → target, the likelihood of observing positive transfers steadily increase across all training setups. For example, the probability of observing positive transfers increases from 88% to 97% when the intermediate thresholds increase from 0.01 to 0.04. These results indicate a transitive behavior between positive transfers.

## 4 Task Selection for Unseen Target Tasks

Pairwise transfer scores are not always available for a new target task. We introduce TASKSHOP to estimate transfer from a source task in TASKWEB to an unseen target task with only a small number of examples (Figure 2). Then, we perform task selection in two settings: a single-task setup where we identify a helpful source task, and a multi-task setup where we locate a set of helpful source tasks.

### 4.1 TASKSHOP: Selecting Helpful Tasks

The objective of task selection in a single-task setup is to predict the benefit of initializing a model on a source task for learning a target task. We introduce a new method TASKSHOP which uses pairwise transfer scores to estimate the transfer from source tasks in TASKWEB to an unseen target task.

**Setup.** Given a source task $s \in S$ and an unseen target task $t$, we seek to predict the transferability of $s$ to $t$. We assume access to pairwise transfer scores between $s$ and other source tasks $S\backslash\{s\}$. Meanwhile, we have a small number of examples ($n \leq 32$) but no pairwise transfer scores for $t$.

**Overview.** Our method searches over paths from $s$ to $t$ via a set of pivot tasks in TASKWEB where each pivot $p$ forms a path $s \rightarrow p \rightarrow t$, and averages their scores to estimate $s \rightarrow t$. It builds upon our previous observation that the strengths of $s \rightarrow p$ and $p \rightarrow t$ help us estimate the strength of $s \rightarrow t$.

**Method.** The TASKSHOP method is summarized in Equation 4.1. Given a pivot task $p \in S\backslash\{s\}$ for which transfer $s \rightarrow p$ is already known, we first use an off-the-shelf task selection method $F$ to obtain $F(p \rightarrow t)$. $F$ can be any method that only uses a small number of task examples. Then, we find the

pairwise transfer score $T(s \rightarrow p)$ from TASKWEB, and average the two scores. We repeat this process over all pivot tasks $p \in S\backslash\{s\}$ and average the resulting scores. Finally, we linearly interpolate our estimate with a direct estimate $F(s \rightarrow t)$ using a hyperparameter $\lambda$ tuned on a held-out task.

$$\text{TS}(s,t) = \lambda \cdot \frac{1}{\|S\backslash\{s\}\|} \sum_{p \in S\backslash\{s\}} \frac{T(s \rightarrow p) + F(p \rightarrow t)}{2}$$
$$+ (1-\lambda) \cdot F(s \rightarrow t)$$
$$(1)$$

**TASKSHOP is directional.** One interesting feature of TASKSHOP is its *directionality*—our predictions for $A \rightarrow B$ differs from $B \rightarrow A$. Our method deviates from conventional techniques that use task embeddings and select tasks using cosine similarities, which results in symmetric predictions. Hence our method is more aligned with the non-commutative property observed in Section 3.3.

**TASKSHOP is modular.** Another feature of TASKSHOP is its *modularity* since any task selection method that only uses a small number of target examples can be used for $F$. Likewise, we utilize recent methods that only use a small number of target task examples, thereby excluding methods that require the fine-tuned model or the full training set. Specifically, we use Retrieval-of-Experts (RoE) from Jang et al. (2023) and the LLM similarity method from Paranjape et al. (2023) for $F$.

### 4.2 Extension to Multi-Task Selection

While choosing a single, appropriate source task is beneficial for learning a target task (Vu et al., 2020, 2022), it has also been observed that using multiple source tasks provides additional benefits (Asai et al., 2022). Hence we extend task selection from a single-task to a multi-task setup.

Given a target task $t$ and a task selection method, we first select the top-$k$ highest scoring source tasks $S_k = \{s_1, ..., s_k\}$ for $t$. Here, the task selection method can be TASKSHOP or other methods. We then randomly sample $n$ prompted examples from each task, resulting in a small training set of $kn$ examples. Table 6 in the appendix shows examples of top-5 tasks selected by TASKSHOP with $F$=RoE.

## 5 Experiments and Results

### 5.1 Single-Task Selection

**Comparisons.** We compare to **Retrieval-of-Experts (RoE)** from Jang et al. (2023) and **LLM-similarity** in Paranjape et al. (2023). For Retrieval-of-Experts, we take 100 examples of the source

| | Method | NLI/Entailment | Paraphrase | Commonsense | Sentiment | QA | Semantics | **Mean** |
|---|---|---|---|---|---|---|---|---|
| **NDCG** | LLM similarity | 54.75 | 47.01 | 63.14 | 65.71 | 41.96 | 56.07 | 56.69 |
| | Retrieval-of-Experts | 66.53 | 49.19 | 65.7 | 78.21 | 84.46 | 54.33 | 64.52 |
| | Ours: TASKSHOP$_{LLM}$ | 54.12 | **52.9** | 67.26 | 71.38 | 51.12 | **56.48** | 59.69 |
| | Ours: TASKSHOP$_{RoE}$ | **75.14** | 49.29 | **79.49** | 80.53 | 85.74 | 54.22 | **71.54** ($\uparrow$) |
| **Regret@5** | LLM similarity | 3.31 | 1.84 | 6.92 | 0.56 | 3.79 | **0.78** | 3.67 |
| | Retrieval-of-Experts | 4.79 | 1.38 | 6.83 | 0.14 | 4.26 | 1.84 | 4.11 |
| | Ours: TASKSHOP$_{LLM}$ | **3.31** | **0.85** | 4.37 | 0.22 | 3.79 | 0.86 | 2.73 |
| | Ours: TASKSHOP$_{RoE}$ | 3.51 | 1.35 | **3.76** | **0.04** | **2.22** | 1.67 | **2.66** ($\downarrow$) |

Table 2: Results of task selection experiments. We use TASKWEB to evaluate TASKSHOP and two task selection methods that only use target examples : LLM similarity (Paranjape et al., 2023) and RoE (Jang et al., 2023). TASKSHOP $_{LLM}$ uses $F$ = LLM-similarity and TASKSHOP $_{RoE}$ uses $F$ = RoE in Equation 4.1. TASKSHOP $_{RoE}$ exhibits the best performance in task selection both in terms of the overall ranking (NDCG) and top-5 precision (Regret@5). Note that a higher score is better for NDCG (above) and a lower score is better for Regret@5 (below).

task and 32 examples of the target task and compute the similarity between text embeddings of the prompts. For LLM-similarity, we input a prompt to text-davinci-003 (Ouyang et al., 2022) to assign probability scores to whether the two tasks are similar or not. For TASKSHOP, we use RoE and LLM-similarity for $F$ in Equation 4.1. More details are provided in Section A.1 in the appendix.

**Metrics.** To evaluate task selection, we use two metrics: normalized discounted cumulative gain (NDCG) and Regret@$k$, following Poth et al. (2021). We use NDCG to evaluate the overall ranking, and Regret@$k$ to measure the performance drop of the predicted top-$k$ source tasks from the actual top-$k$ source tasks. We evaluate task selection for all tasks in our setup grouped by categories in Table 1, and use TASKWEB for the gold labels.

**Experimental Setup.** While we use target tasks from TASKWEB to use their transfer scores as labels, we wish to simulate a scenario in which there are only 32 examples for each target. Therefore we perform our experiments in a leave-one-out setup, where for each experiment we assume access to pairwise scores amongst our set of tasks except for the given target task. In this way, we maintain the assumption that only a small number of examples of the target task are available during evaluation.

**Results.** Table 2 reports our results. Combining pairwise transfer scores with LLM and RoE improves both NDCG and Regret@5 compared to their base methods, with the best gains from RoE. We hypothesize that the improvement occurs because the pairwise transfer scores capture the transferability between each source task and the set of tasks textually similar to the target task. Due to

transitive behavior between positive task transfers, these transfer scores would provide additional information about the transferability from the helpful source tasks to the target. Moreover, our method considers the direction of the pairwise transfer unlike the other methods, thereby better accounting for the non-commutativity observed in Section 3.3.

## 5.2 Multi-Task Selection

We now investigate whether TASKSHOP can also be used to select multiple source tasks that collectively improve target task performance.

**Comparisons.** We use the following baselines. **T0-3B** has the same architecture as T5-3B but trained on millions of examples spanning 35 different tasks (Sanh et al., 2022). **T5-3B + most similar** is the LM-adapted T5-3B (Lester et al., 2021) trained on a handpicked, similar source task from the same category as each target task. **T5-3B + all tasks** is the LM-adapted T5-3B trained with samples from all 22 tasks from TASKWEB except each target task in a leave-one-out setup.

We then train T5-3B models on small training sets sampled from the five highest-scoring source tasks based on the following task selection methods: **Retrieval-of-Experts** from (Jang et al., 2023), **LLM-similarity** from (Paranjape et al., 2023) and **TASKSHOP $_{RoE}$** with $F$ = RoE in Equation 4.1.

Finally, we consider the case where **TASKWEB** scores for the target task are available and select the five highest-scoring source tasks for each target. We train T5-3B on samples from these tasks.

**Training Setup.** Given a target task $t$ and a task selection method, we first select the five highest-scoring source tasks $s_1, ..., s_5$ for $t$. We then randomly sample 2,000 prompted examples from each

| Method | ANLI-R1 | ANLI-R2 | ANLI-R3 | CB | COPA | Hellasw. | RTE | StoryC. | WiC | Winogr. | WSC | Mean |
|---|---|---|---|---|---|---|---|---|---|---|---|---|
| T0-3B | 35.62 | 33.36 | 33.10 | 62.20 | 75.50 | 27.30 | 61.87 | 85.13 | 50.88 | 50.65 | **66.02** | 52.88 |
| T5-3B + most similar | **44.50** | **37.42** | 39.61 | 79.07 | 81.42 | 41.46 | 72.83 | 93.73 | 50.86 | 52.83 | 36.54 | 57.30 |
| T5-3B + all tasks | 41.49 | 35.32 | 39.61 | 79.96 | 82.08 | 39.73 | 74.95 | 91.93 | 52.93 | **57.35** | 44.44 | 58.16 |
| Retrieval-of-Experts* | 38.38 | 35.44 | 41.24 | 75.2 | 83.17 | 41.86 | 65.08 | 94.04 | **53.22** | 50.09 | 44.76 | 56.59 |
| LLM-similarity◇ | 39.91 | 34.74 | 38.84 | 81.65 | 80.91 | 40.85 | **78.2** | 93.96 | 51.35 | 52.26 | 55.02 | 58.88 |
| Ours: TASKSHOP$_{\text{RoE}}$ | 42.86 | 36.15 | **41.41** | 84.52 | 86.08 | 41.94 | 76.73 | 94.04 | 51.49 | 53.0 | 59.4 | **60.69** |
| Ours: TASKWEB † | 40.16 | 36.15 | 42.15 | 82.24 | 85.25 | 43.73 | 77.71 | 92.69 | 50.75 | 55.84 | 62.82 | 60.86 |

Table 3: Results of multi-task learning experiments. We perform all evaluations in zero-shot settings, meaning that we do not fit the model parameters to the target task - however, we still assume access to a small number of labeled examples of the target. We average results over multiple prompts. The first group corresponds to our baselines, the second group corresponds to two existing task selection methods, as well as TASKSHOP without access to TASKWEB scores for the target task (but access to TASKWEB scores between other tasks), and the third group uses TASKWEB scores for the target task to select source tasks. ⋆ is from Jang et al. (2023) and ◇ is from Paranjape et al. (2023). † has access to TASKWEB scores directly to the target task. All methods below the dotted line use the top-5 scoring source tasks to build multi-task training sets, while the three above utilize different numbers of source tasks.

task and randomly shuffle all examples to create a multitask training set. For the T5-3B most similar baseline, we sample 10,000 examples of the similar task in the same category in order to ensure that the size of the training set is the same as the size of the multitask training sets in our other experiments. Meanwhile, for the T5-3B + all tasks baseline, we select 21 tasks except the target and use 2,000 examples from each task. We provide more training details in the appendix.

As it is costly to compute pairwise transfer scores with bigger language models, we use TASKWEB scores from T5-large. This is based on our observation that models with similar architectures and adaptation methods share more similar transferabilities (Section 3.2). We hypothesize that T5-large can learn the complexities of our source tasks and represent their transferabilities—this is supported by how both our T5-large transfers and T5-3B expert models in Jang et al. (2023) found CosmosQA and SocialIQA to be great source tasks.

**Evaluation setup.** We use the same set of evaluation tasks used by Jang et al. (2023). For ANLI-R1/R2 which are not included in TASKWEB, we apply the subset of tasks chosen for ANLI-R3 for the upper baseline. Meanwhile, for the Story Cloze task which is not included in TASKWEB due to its lack of training set, we use a subset of five tasks with the best transfer scores for the upper baseline. For each target task, we perform the evaluation in a leave-one-out setup by removing the target task from TASKWEB along with its scores. This is to maximize the number of available source tasks

while ensuring that the target task is unseen in our setup. By doing so, we simulate using TASKSHOP and TASKWEB across various categories of target tasks with access only to their examples ($n \leq 32$). We perform all evaluations in a zero-shot setting.

**Results.** Table 3 summarizes the results of our experiments. The middle section details the performances of task selection methods that assume no access to pairwise transfer scores to the target. Two out of three methods improve target task performance compared to all baselines. Most notably, TASKSHOP outperforms both baselines as well as other task selection methods, improving by 14.7% over T0-3B and by 4.3% over our strongest baseline while using a small portion of the training set.

Finally, we observe that using the top-5 source tasks for each target according to TASKWEB consistently improves target performance. Our results support previous observations that using smaller multi-task training sets with a more careful task selection strategy can improve target performance (Pruksachatkun et al., 2020; Chan et al., 2022).

### 5.3 Discussion

The results of our experiments indicate that single-task transfer metrics can help improve multi-task transfers. We perform further experiments to support this hypothesis and address three questions.

**How many source tasks do we need?** We investigate whether different numbers of source tasks in the training set affect target task performance. To this end, we train T5-3B on training sets with top-1, 3, 10 and 21 source tasks in addition to five tasks.

| Method | ANLI-R1 | ANLI-R2 | ANLI-R3 | CB | COPA | Hellasw. | RTE | StoryC | WiC | Winogr. | WSC | Mean |
|---|---|---|---|---|---|---|---|---|---|---|---|---|
| Top-1 | 40.83 | 34.53 | 38.08 | 75.0 | 80.08 | 28.56 | 70.49 | 89.68 | 50.74 | 52.6 | 36.54 | 54.28 |
| Top-3 | 41.78 | **36.54** | 40.86 | 79.46 | **86.16** | **45.54** | 70.54 | 89.66 | 51.32 | 52.61 | 54.81 | 59.03 |
| Top-5 | **42.86** | 36.15 | **41.41** | **84.52** | 86.08 | 41.94 | 76.73 | **94.04** | 51.49 | 53.0 | **59.4** | **60.69** |
| Top-10 | 40.58 | 35.17 | 38.88 | 75.6 | 84.92 | 42.24 | **78.65** | 93.99 | 51.41 | 52.54 | 58.97 | 59.36 |
| Top-21 | 41.49 | 35.32 | 39.61 | 79.96 | 82.08 | 39.73 | 74.95 | 91.93 | **52.93** | **57.35** | 44.44 | 58.16 |

Table 4: Results of choosing different numbers of source tasks for multi-task learning with TASKSHOP ᵣₒₑ. For each target task, the highest scoring setup is **bolded**. Results for top-5 are taken from TASKSHOP$_{RoE}$ in Table 3.

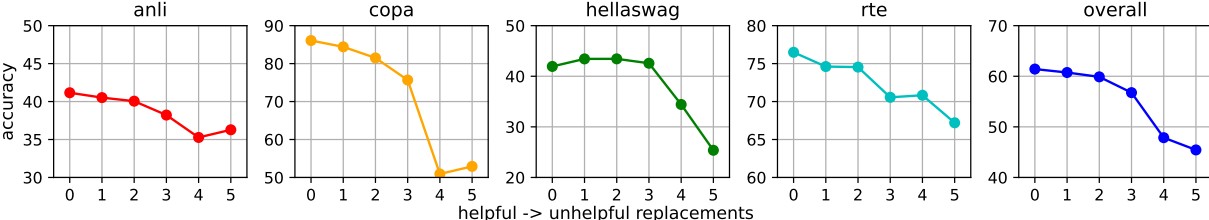

Figure 4: Variations in the zero-shot target performance as the top-5 source tasks for each target are incrementally replaced by the bottom-5 source tasks according to TASKWEB while maintaining the size of the training set.

| Method | ANLI | COPA | Hellasw. | Mean |
|---|---|---|---|---|
| Random | 35.25 | 72.58 | 29.64 | 50.51 |
| Bottom-5 w/ TASKSHOP | 34.41 | 55.92 | 25.01 | 47.13 |
| Bottom-5 w/ TASKWEB | 34.72 | 52.92 | 25.37 | 46.25 |

Table 5: Results of choosing random and worst sets of tasks according to TASKSHOP and TASKWEB for three example target tasks, as well as the mean over all target tasks. Table 9 in the appendix provides the full results.

Table 4 shows the results. We observe that most target tasks achieve performance improvements from training on 3 to 5 source tasks. Using five source tasks results in the best overall performance and ranks first or second across most targets. Meanwhile, using ten source tasks results in a worse overall performance. The performance drops considerably when 21 tasks are used. According to our results, most targets only require a careful selection of three to five source tasks except several tasks such as Winogrande. Our findings differ from previous work which finds performance to scale with the number of tasks (Sanh et al., 2022; Wei et al., 2022a; Wang et al., 2022b) because while they add tasks in a target-agnostic manner, we add helpful source tasks based on the target task.

**Do our methods identify both helpful and unhelpful source tasks?** We demonstrate that our methods can also identify *unhelpful* tasks in multi-task settings. To this end, we pick the bottom-5 source tasks for each target with TASKSHOP and TASKWEB, as well as five random source tasks.

Table 5 summarizes the results. A random set of source tasks underperforms the T0-3B baseline, and the bottom-5 tasks from TASKSHOP further observes decreases in 3.4 accuracy points on average. Finally, the bottom-5 tasks based on TASKWEB results in similarly low performances. These results indicate that negative pairwise transfers between source and target tasks impact multi-task learning.

**What happens if we mix helpful and unhelpful source tasks?** While grouping helpful sources improves target performance and vice versa, it is unclear what happens in between. To address this, we experiment with different proportions of helpful tasks and measure the target task performance. We repeat this process over four target tasks in our evaluation setup—ANLI (R3), COPA, HellaSwag and RTE. For each task, we start with the top-5 tasks according to TASKWEB and replace a task with a bottom-5 task until all top-5 tasks are replaced. We perform the same evaluations as Tables 3, 4 and 5.

Figure 4 visualizes the results. As each helpful source task is replaced with an unhelpful source task, the target performance decreases across all four tasks. However, there are several instances where such replacement *increases* performance, as can be seen from 0→1 in HellaSwag and 4→5 in ANLI. These results indicate that while pairwise transferability between the source and target heavily impacts target performance during multi-task learning, other factors such as negative interference between the source tasks may also be involved, which is an interesting direction for future work.

## 6 Conclusion

In this work, we investigate how using prior knowledge of task relationships quantified via pairwise task transfer aids selecting helpful source tasks for multi-task NLP. We build TASKWEB, a benchmark and repository of pairwise task transfers across different tasks, models and adaptation methods in NLP. Based on our analysis of TASKWEB, we propose TASKSHOP, our method for selecting helpful source tasks for a new target task. We show that TASKSHOP outperforms existing methods in choosing helpful source tasks for different target tasks. Moreover, we use TASKSHOP and TASKWEB to build small multi-task training sets and outperform other methods that use much larger training sets.

## 7 Limitations

Our work contains several limitations. First, our set of tasks does not constitute the entirety of NLP tasks. While we use 22 NLP tasks that are representative enough to cover various types of reasoning, we do not include long-form tasks (e.g., summarization, LFQA) or domain-specific tasks (e.g., law, medicine) to facilitate experiments across various model architectures such as encoder-only models. In order to add entirely new forms of task to TASKWEB, one would have to compute pairwise transfer scores between the new task and other tasks in TASKWEB. If the model is known beforehand, this would require $\|T\|$ iterations of fine-tuning with 1,000 examples where $T$ is the set of tasks in TASKWEB. On the other hand, if the model is not known beforehand, this would require $\|M\| \times \|T\|$ iterations where $M$ is the set of models used in TASKWEB.

Moreover, our datasets are in English and we do not incorporate multilinguality in our experiments. Second, our work focuses on models with at most three billion parameters. Our finding may not be directly applicable to models with orders of magnitude more parameters considering factors such as emergence (Wei et al., 2022b), which can be explored in future work. Third, we perform our multi-task finetuning experiments by uniformly sampling 2,000 examples from each source task following the style of Wang et al. (2022b). Therefore, different behavior may arise when other sampling strategies are used. Finally, recent work shows the effectiveness of using diverse instruction-output pairs which do not necessarily have clear boundaries as our tasks do (Ouyang et al., 2022; Wang et al., 2022a, 2023). Recently, Wang et al. 2023 report that large language models finetuned on specific instruction datasets perform better on related target tasks, which is closely related to our findings. Future work could extend our approach to setups without clear boundaries between tasks and explore ways to perform target-specific instruction tuning. Considering these limitations, we encourage the NLP community to contribute to quantifying the transferabilities between different language tasks.

## Ethics Statement

TASKWEB is based on a set of representative NLP tasks that have widely been used in the NLP community. While this work explores pairwise task transfer and multi-task finetuning using non-harmful datasets, an adversary could potentially misuse our approach to build another version of TASKWEB containing harmful tasks and quickly train models specifically for malicious target tasks. Hence we emphasize the importance of monitoring the content of tasks newly added to TASKWEB.

## Acknowledgements

We thank members of the H2Lab and UW NLP for their discussion and constructive feedback. This work was funded in part by the DARPA MCS program through NIWC Pacific (N66001-19-2-4031), NSF IIS-2044660, and gifts from AI2. Joongwon Kim is supported by the National Science Foundation Graduate Research Fellowship under Grant No. DGE-2140004. Akari Asai is funded by the IBM PhD Fellowship.

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

## A  Appendix

### A.1  More Experimental Details

**Full list of the datasets.**   Table 5 presents the complete list of the 22 tasks studied in TASKWEB, along with references to the original papers.

**Pairwise Task Transfer Metric**   For a source $s$ and target $t$, evaluation function $p$, model $m_t$ tuned on $t$ and a model $m_{s \rightarrow t}$ tuned from $s$ to $t$,

$$\text{PC}(s,t) \underset{m \in M}{\propto} \frac{p(m_{s \rightarrow t}) - p(m_t)}{p(m_t)}$$

$$\text{PM}(s,t) \underset{m \in M}{\propto} \mathbb{1}\left(p(m_{s \rightarrow t}) > p(m_t)\right)$$

PC refers to the average percentage change of the model performance across all random seeds, and PM refers to the proportion of models that resulted in a positive transfer across all random seeds.

**Implementation Details of Task Selection.**   For Retrieval-of-Experts, we use a similar implementation by taking 100 examples of the source task and 32 examples of the target task and computing the similarity between text embeddings of the prompts. We use PromptSource (Bach et al., 2022) to extract prompts and Sentence Transformers (Reimers and Gurevych, 2019) to obtain text embeddings.

For LLM-similarity, we write a prompt that contains several pairs of tasks not used in our setup, where each pair has 1) an example of each task, and 2) an answer noting whether the two tasks are similar or not. Then, for each source-target pair, we pass the prompt prepended to source and target examples to text-davinci-003 (Ouyang et al., 2022). We use the ratio of the log probabilities of the answers "yes" and "no" to assign a score between the source and target tasks.

**Multi-Task Finetuning Details.**   We construct our multi-task training set by randomly selecting 2,000 examples with prompts from each task. For our **T5-3B + all tasks** baseline we choose all 21 tasks in TASKWEB apart from the target task, resulting in 42,000 examples. For all other methods (**Retrieval-of-Experts, LLM-similarity, TASKSHOP$_{\text{RoE}}$, TASKWEB**), we choose the five highest-scoring tasks according to each method, resulting in 10,000 examples. Then, we fully fine-tune LM-adapted T5-3B on our training set for five epochs, with an Adam Optimizer using a learning rate of $1e$-4 and batch sizes ranging from 4 to 16 depending on the maximum length of each dataset.

| Target | Selected Tasks |
|--------|----------------|
| ANLI | RTE, CB, SNLI, CsmsQA, Soc.IQA |
| CB | ANLI, CsmsQA, Soc.IQA, WSC, SNLI |
| COPA | CsmsQA, Soc.IQA, Winogr., Hellasw., PIQA |
| Hellasw. | PIQA, CsmsQA, Soc.IQA, Winogr., COPA |
| RTE | ANLI, QNLI, Soc.IQA, MRPC, SQuADv2 |
| StoryC. | CsmsQA, COPA, Soc.IQA, Hellasw., Winogr. |
| WiC | PIQA, MRPC, ANLI, Hellasw., Soc.IQA |
| Winogr. | Soc.IQA, CsmsQA, PIQA, COPA, WSC |
| WSC | Winogr., ANLI, Soc.IQA, WIC, RTE |

Table 6: Top-5 source tasks selected using TASKSHOP.

| Target | Selected Tasks |
|--------|----------------|
| ANLI | CsmsQA, BoolQ, SNLI, Rot.Tom, RTE |
| CB | ANLI, BoolQ, SNLI, Rot.Tom, SciTail |
| COPA | CsmsQA, Winogr., SciTail, PIQA, Soc.IQA |
| Hellasw. | CsmsQA, Soc.IQA, PIQA, RTE, Rot.Tom |
| RTE | ANLI, CsmsQA, Winogr., SQuADv2, Soc.IQA |
| StoryC. | CsmsQA, Soc.IQA, PIQA, Winogr., Rot.Tom |
| WiC | QNLI, MRPC, SNLI, RTE, ANLI |
| Winogr. | SQuADv2, Soc.IQA, CsmsQA, ANLI, Quartz |
| WSC | ANLI, QNLI, QQP, Soc.IQA, SNLI |

Table 7: Top-5 source tasks selected using TASKWEB.

### A.2  More Pair-wise Transfer Results

**Full results.**   Figure 5 displays pairwise transfer scores for all tasks in TASKWEB averaged over training setups. Scores for individual setups are shown in Figure 6 (T5-large finetune), Figure 7 (T5-base finetune), Figure 8 (RoBERTa-base finetune), Figure 9 (GPT2-medium finetune), Figure 10 (T5-base Adapters), Figure 11 (T5-base BitFit) and Figure 12 (T5-small finetune).

**Commutativity results.**   Figure 13 shows the commutativity experiment results.

**Transitivity results.**   Figure 14 shows the experimental results of the transitivity analysis for all setups in our experiments.

### A.3  More Multi-Task Selection Results

**Tasks chosen for the multi-task setup.**   Tables 6 and 7 list the top-5 (left to right) source tasks chosen for our multi-task setup using TASKSHOP and TASKWEB, respectively.

**Bottom-5 and random-5 full results.**   Table 9 presents the evaluation results for the bottom-5 source tasks selected with TASKSHOP and TASKWEB as summarized in Table 5, as well as five random source tasks.

*datasets used in our experiments*

ANLI (Nie et al., 2020), BoolQ (Clark et al., 2019), CB (de Marneffe et al., 2019), COPA (Gordon et al., 2012), CosmosQA (Huang et al., 2019), HellaSwag (Zellers et al., 2019), IMDB (Maas et al., 2011), MRPC (Dolan and Brockett, 2005), PIQA (Bisk et al., 2020), QNLI (Demszky et al., 2018), QQP (Wang et al., 2017), QuaRTz (Tafjord et al., 2019), Rotten Tomatoes (Pang et al., 2002), RTE (Candela et al., 2006; Bar-Haim et al., 2006; Giampiccolo et al., 2007; Bentivogli et al., 2009), SciTail (Khot et al., 2018), SNLI (Bowman et al., 2015), SocialIQA (Sap et al., 2019), SQuAD2.0 (Rajpurkar et al., 2018), Story Cloze (Mostafazadeh et al., 2017), STSB (Cer et al., 2017), WiC (Pilehvar and Camacho-Collados, 2019), Winogrande (Sakaguchi et al., 2020), WSC (Levesque et al., 2012)

Table 8: References for datasets used in our experiments.

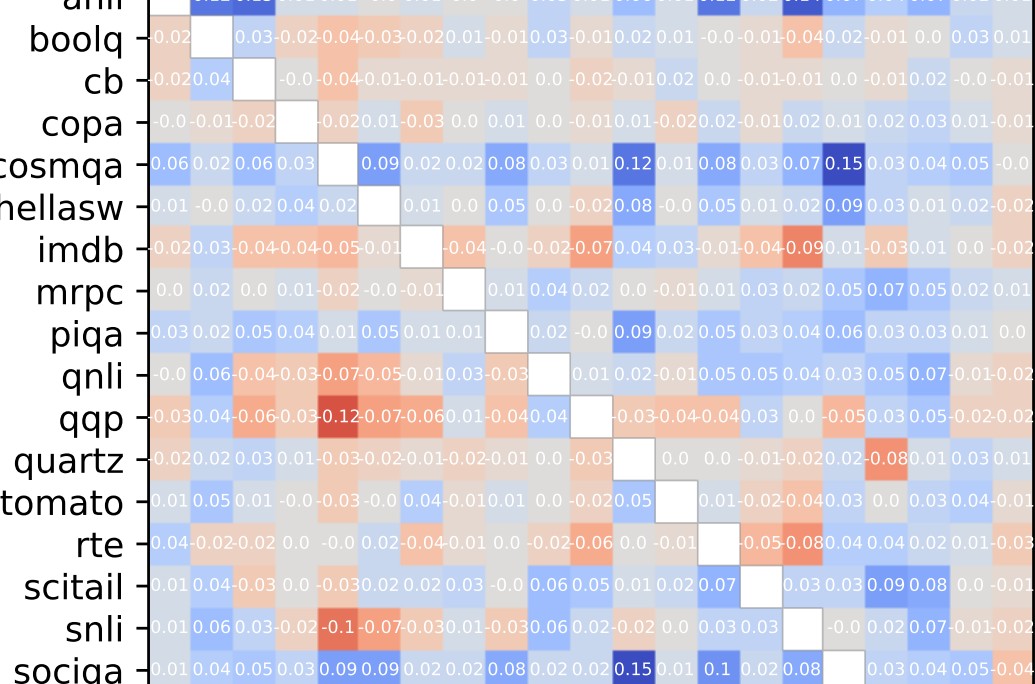

Figure 5: Visualization of pairwise transfer between 22 different NLP tasks, averaged over our training setups. We display the actual transfer scores, with positive transfers in blue and negative transfers in red.

| Method | ANLI-R1 | ANLI-R2 | ANLI-R3 | CB | COPA | Hellasw. | RTE | StoryC | WiC | Winogr. | WSC | Mean |
|---|---|---|---|---|---|---|---|---|---|---|---|---|
| Random | 34.35 | 35.29 | 36.12 | 65.67 | 72.58 | 29.64 | 73.69 | 55.84 | 49.53 | 51.03 | 51.92 | 50.51 |
| Bottom-5 w/ TASKSHOP | 33.39 | 34.21 | 35.63 | 67.76 | 55.92 | 25.01 | 62.57 | 59.42 | 50.33 | 50.45 | 43.69 | 47.13 |
| Bottom-5 w/ TASKWEB | 34.33 | 33.56 | 36.28 | 47.02 | 52.92 | 25.37 | 67.2 | 57.3 | 50.05 | 50.1 | 54.59 | 46.25 |

Table 9: Results of choosing random and worst sets of tasks according to TASKSHOP and TASKWEB. Refer to the third row in Table 4 for target task performances with the top-5 source tasks selected by TASKSHOP.

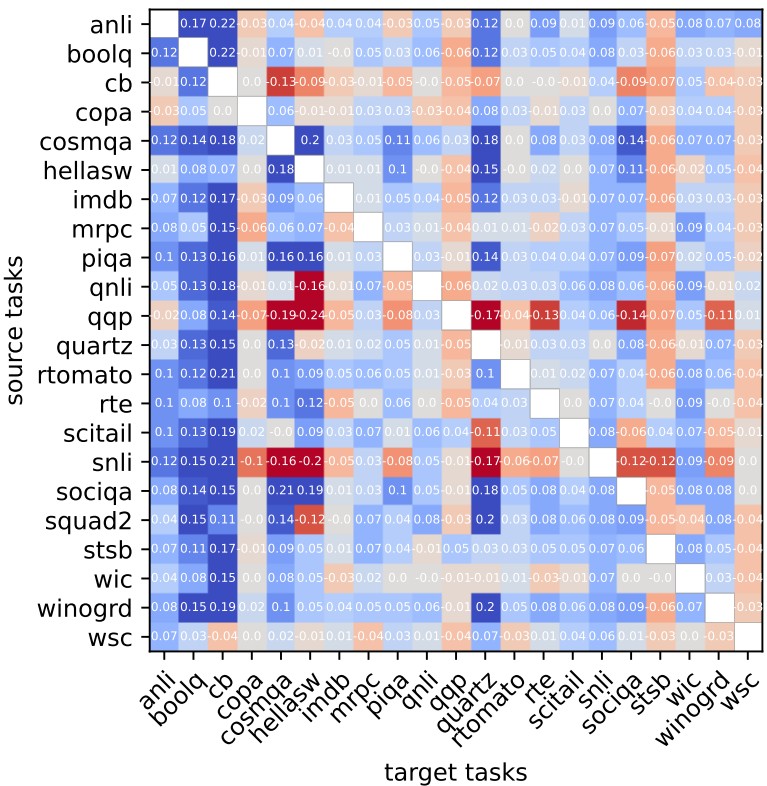

Figure 6: Visualization of pairwise transfer between 22 different NLP tasks for T5-large ([Raffel et al., 2020](#)) finetune. We display the actual transfer scores, with positive transfers in blue and negative transfers in red.

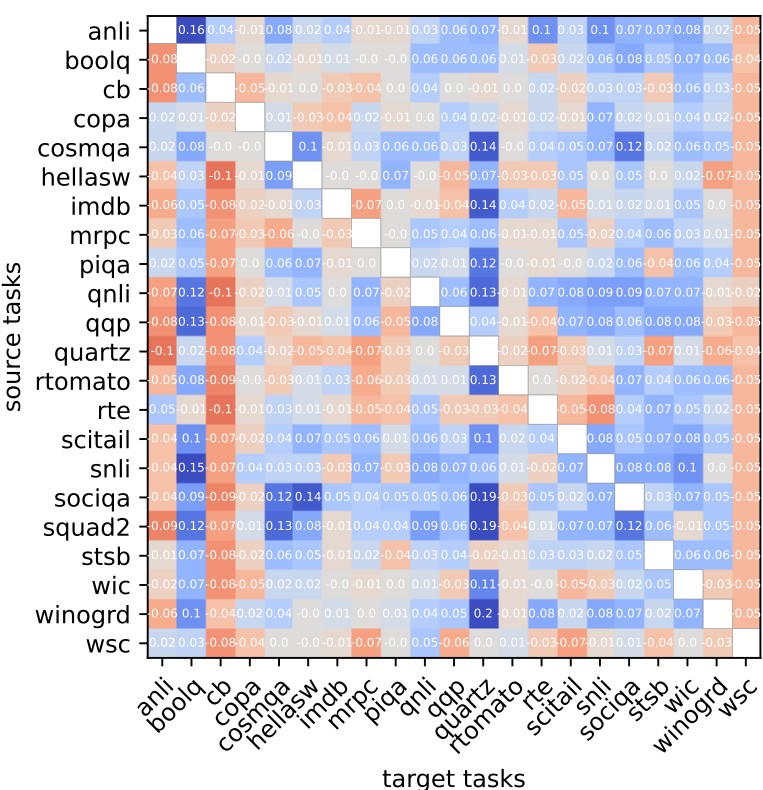

Figure 7: Visualization of pairwise transfer between 22 different NLP tasks for T5-base finetune. We display the actual transfer scores, with positive transfers in blue and negative transfers in red.

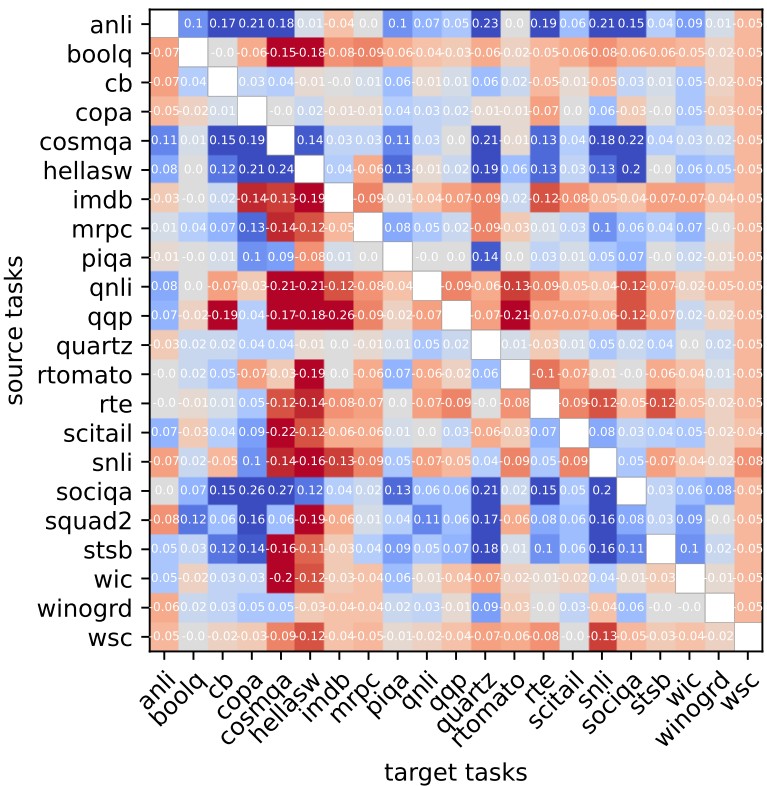

Figure 8: Visualization of pairwise transfer between 22 different NLP tasks for RoBERTa-base (Liu et al., 2019) finetune. We display the actual transfer scores, with positive transfers in blue and negative transfers in red.

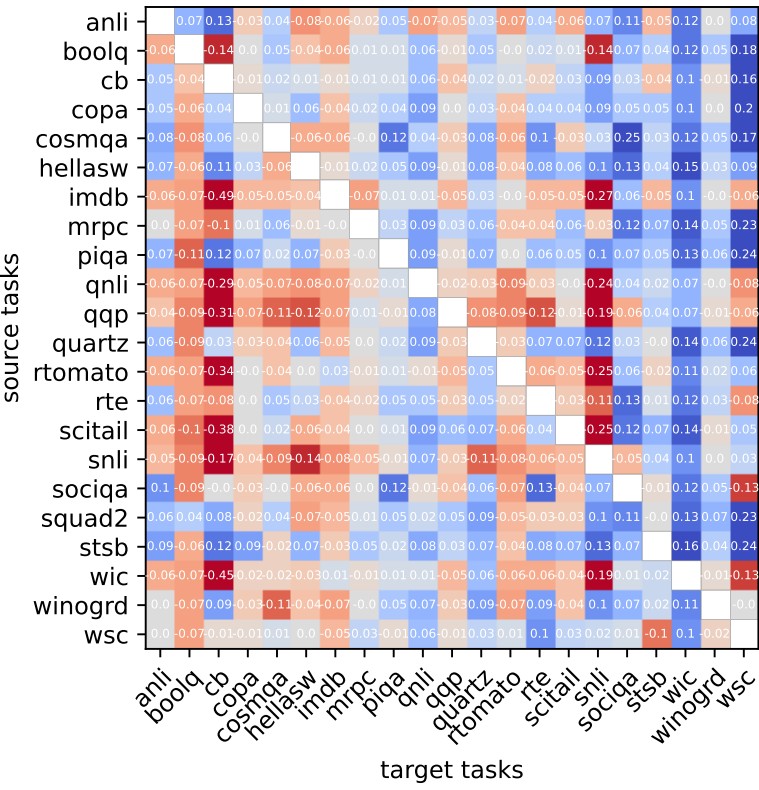

Figure 9: Visualization of pairwise transfer between 22 different NLP tasks for GPT-2 medium (Radford et al., 2019) finetune. We display the actual transfer scores, with positive transfers in blue and negative transfers in red.

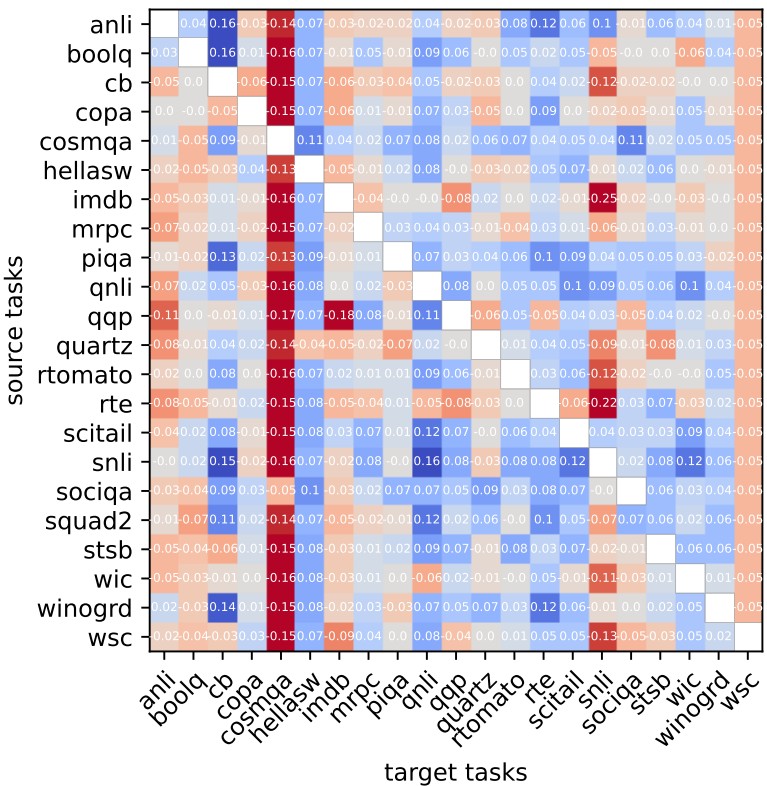

Figure 10: Visualization of pairwise transfer between 22 different NLP tasks for T5-base adapters (Houlsby et al., 2019). We display the actual transfer scores, with positive transfers in blue and negative transfers in red.

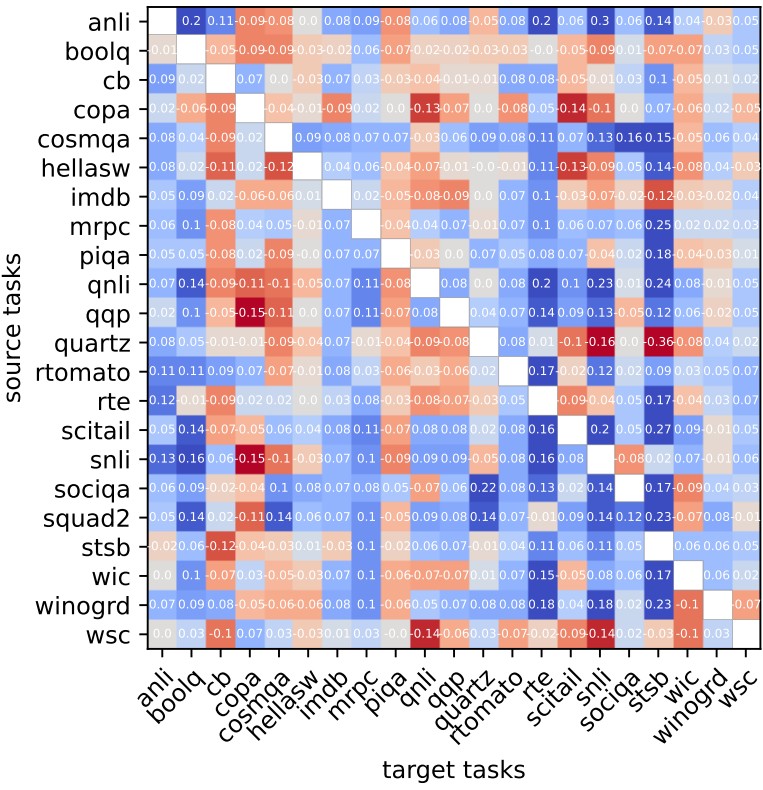

Figure 11: Visualization of pairwise transfer between 22 different NLP tasks for T5-base BitFit (Zaken et al., 2022). We display the actual transfer scores, with positive transfers in blue and negative transfers in red.

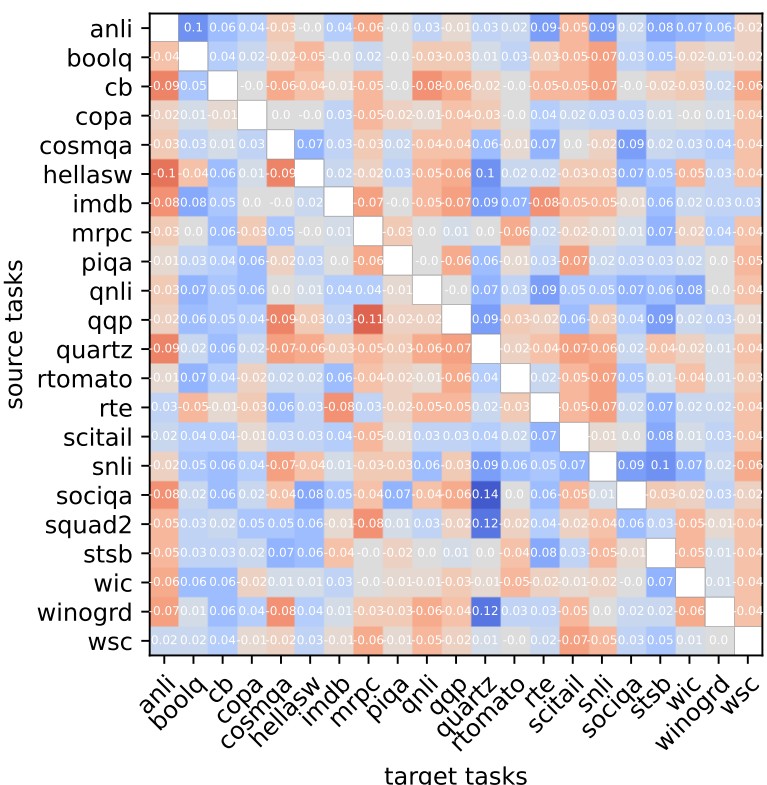

Figure 12: Visualization of pairwise transfer between 22 different NLP tasks for T5-small finetune. We display the actual transfer scores, with positive transfers in blue and negative transfers in red.

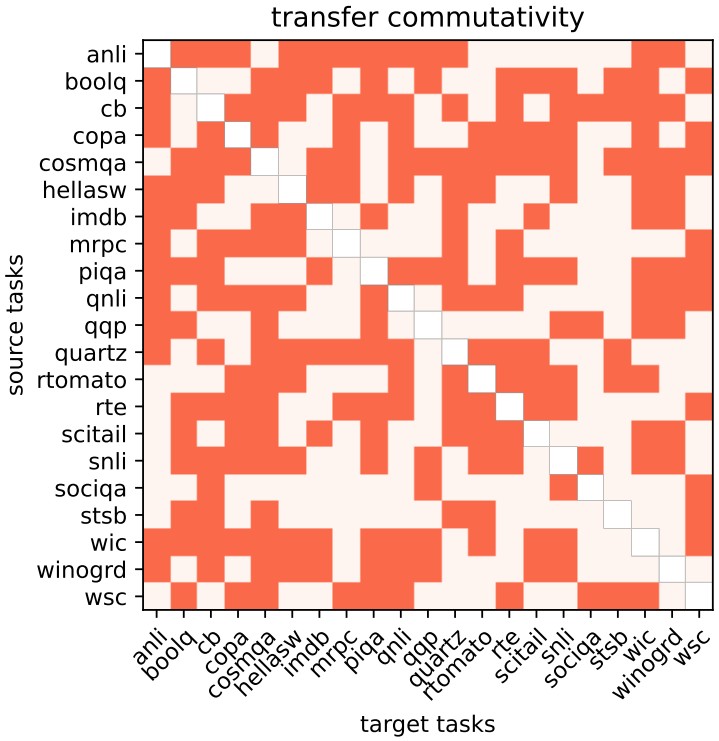

Figure 13: Visualization of commutativity between all tasks in our pairwise transfer setup. The white color indicates that transfers in both directions share the same signs, and the orange color indicates opposite signs.

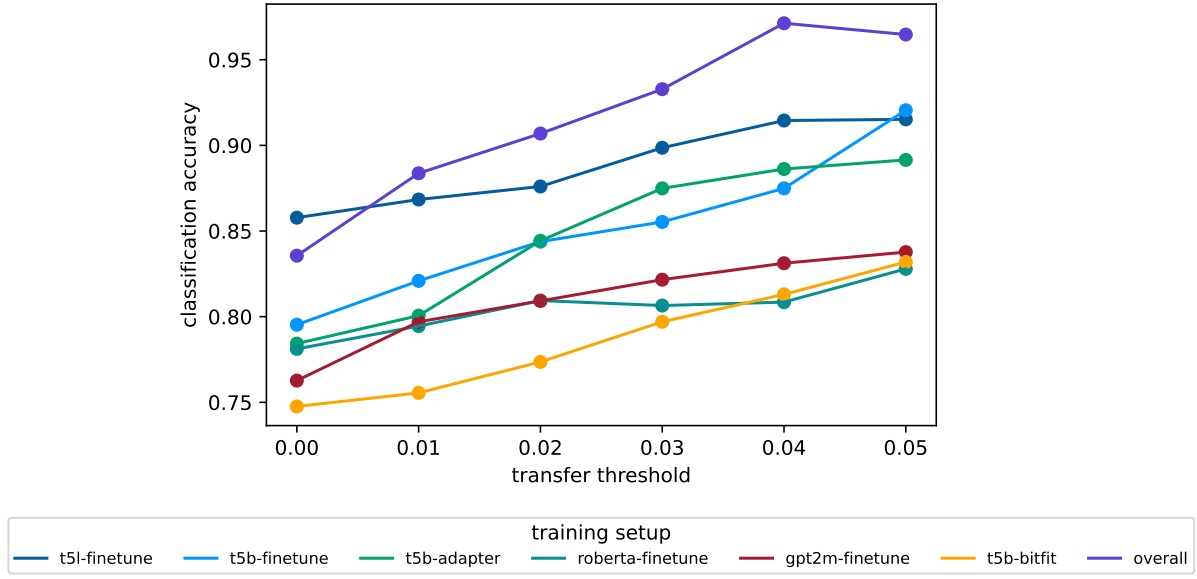

Figure 14: Results for Figure 3 (right) but for all setups in TASKWEB, with the probability of identifying positive source → target transfers as the minimum threshold for (source → pivot, pivot → target) transfers is increased.