# OpenReview forum: "TaskWeb: Selecting Better Source Tasks for Multi-task NLP"
_EMNLP/2023/Conference — EMNLP 2023 Main_

### Official Review · Reviewer_5RiK · 2023-08-05

**Soundness:** 4

**Excitement:**

4: Strong: This paper deepens the understanding of some phenomenon or lowers the barriers to an existing research direction.

**Paper Topic And Main Contributions:**

This paper focuses on source task selection in transfer learning. Specifically, the paper study whether knowing the pairwise task relationship helps selecting better source tasks. The study comes along with a large scale benchmark and a new method to estimate the usefulness of a source task.

**Questions For The Authors:**

A. When performing multi-task learning, how did you decide the training order? Did you shuffle training data from all tasks or you train with a task first and another?

**Reasons To Accept:**

The paper conducts in-depth investigation on the relationship of source and target tasks, and provide insights on task selection for transfer learning with comprehensive empirical results. The idea is clear and interesting. The authors propose a benchmark TaskWeb for pairwise task transfer, which includes pairwise transfer between all pairs of 22 tasks. Experiments are conducted with three model architectures including encoder-based, decoder-based and encoder decoder-based, which is comprehensive. The observations on the commutativity and transitivity are insightful and valuable to the community. Leveraging the observation that transitivity exists, they propose TaskShop for estimating the source task usefulness. The proposed method shows effectiveness over baselines though experiments on a variety of tasks, demonstrating the significance of the methodology. Lastly, the paper is well-written and easy to follow, which is very appreciable. The claims are clearly stated, supported with empirical evidence.

**Reasons To Reject:**

Overall, the paper is of high quality and the discussion section answers most of my questions. I have one minor concern and put them in the question section.

**Reproducibility:**

5: Could easily reproduce the results.

**Reviewer Confidence:**

3: Pretty sure, but there's a chance I missed something. Although I have a good feel for this area in general, I did not carefully check the paper's details, e.g., the math, experimental design, or novelty.

---

> ### Author Rebuttal · Authors · 2023-08-29
>
> We greatly thank the reviewer for the positive feedback on our submission. More specifically, we appreciate how the reviewer found our idea to be clear and interesting, the experiments to be comprehensive, our observations regarding pairwise transfer results to be insightful, and our experiments to be empirically rigorous.
>
> Regarding the question:
>
> > Q: When performing multi-task learning, how did you decide the training order? Did you shuffle training data from all tasks or you train with a task first and another?
>
> We randomly shuffled training data from all tasks that we chose for each multi-task learning experiment. We will make sure to clarify this point in subsequent iterations of our draft.

---

### Official Review · Reviewer_yFyo · 2023-08-05

**Soundness:** 4

**Excitement:**

4: Strong: This paper deepens the understanding of some phenomenon or lowers the barriers to an existing research direction.

**Paper Topic And Main Contributions:**

This paper first constructs a large-scale benchmark for pairwise task transfers for 22NLP tasks. Then, it introduces a method that predicts the transferability from a source task to a target task. The experimental results are based on single-task and multi-task settings, demonstrating the proposed model's effectiveness.

**Reasons To Accept:**

1. The idea of quantifying the relationship between different NLP tasks is novel and interesting.
2. The paper is well-organized and easy to follow.
3. The experiment design is comprehensive, demonstrating a solid workload.

**Reasons To Reject:**

There have been no tests to determine the statistical significance of the results.

**Reproducibility:**

4: Could mostly reproduce the results, but there may be some variation because of sample variance or minor variations in their interpretation of the protocol or method.

**Reviewer Confidence:**

4: Quite sure. I tried to check the important points carefully. It's unlikely, though conceivable, that I missed something that should affect my ratings.

---

> ### Author Rebuttal · Authors · 2023-08-29
>
> We are excited and thankful to the reviewer for providing us encouraging comments on our submission. More specifically, we appreciate that the reviewer 1) found the idea of quantifying the relationships between different NLP tasks novel and interesting, 2) enjoyed the organization of the paper and 3) considered the experimental design to be comprehensive. As for the statistical significance, we plan to include the associated results in further iterations of the draft.

---

### Official Review · Reviewer_jiqJ · 2023-08-05

**Soundness:** 4

**Excitement:**

3: Ambivalent: It has merits (e.g., it reports state-of-the-art results, the idea is nice), but there are key weaknesses (e.g., it describes incremental work), and it can significantly benefit from another round of revision. However, I won't object to accepting it if my co-reviewers champion it.

**Paper Topic And Main Contributions:**

This paper provides insight on source task selection during transfer learning. Contributions are:
1. The authors investigated pair-wise transferability across 22 popular NLP tasks into a TaskWeb benchmark.
2. They proposed to leverage this benchmark to search source tasks for unseen target tasks. They estimate the transferability of a source task to a target task by (1) the available transfer results between a source task and other tasks in TaskWeb (2) the example similarity between other tasks in TaskWeb and the target task.
3. Through experiments, the authors proved the effectiveness of their source task selection method in both single-task transfer and multi-task transfer settings (the difference is whether the source task is a single one or multiple ones), compared to selection methods that are purely based on example similarity.

**Questions For The Authors:**

A. Why does T have to be directional while F does not? It seems like both the text-embedding method and the LLM method are symmetric when acting as F.

B. Line 442: for ANLI-R1/R2, we apply the subset of tasks chosen for ANLI-R3 for the upper baseline. Can you explain this in more detail? Which baseline is the "upper baseline"? Besides, further explanations may also needed for the story cloze task.

C. In the multi-task transfer setting (table 3), if it is a zero-shot setting, where do the 32 examples come from? Are they from the training set with labels removed? If so, can it be called zero-shot?

**Reasons To Accept:**

1. The TaskWeb benchmark shows pair-wise transferability across 22 popular NLP tasks.
2. The authors summarized useful conclusions from TaskWeb: commutativity does not hold for task transfer but transitivity usually holds.
3. The authors proposed an effective method for selecting source tasks for an unseen target task. The research method of using both existing pair-wise transfer results and example similarity is insightful and could benefit future research with similar situations.
4. The authors showed that by using their task selection method, the selected source tasks are more helpful than pure similarity-based selection baselines or simply using all available source tasks in training. This conclusion holds for both single-task transfer or multi-task transfer.

**Reasons To Reject:**

1. The scalability of this work. Although the research method is insightful, I can hardly imagine a situation where future research does not fall into the categories of TaskWeb (paraphrasing, NLI, sentiment, commonsense QA, reading comprehension, semantic similarity). It is hard to first compute pair-wise transferability of any given set of tasks.
2. It would be better to have a baseline which just uses tasks within the same category as source tasks (i.e., some heuristic baselines).

**Reproducibility:**

4: Could mostly reproduce the results, but there may be some variation because of sample variance or minor variations in their interpretation of the protocol or method.

**Reviewer Confidence:**

4: Quite sure. I tried to check the important points carefully. It's unlikely, though conceivable, that I missed something that should affect my ratings.

---

> ### Author Rebuttal · Authors · 2023-08-29
>
> We would like to thank the reviewer for their thorough evaluation of the paper, and their helpful and insightful comments. More specifically, we appreciate how the reviewer pointed out that 1) we integrate pairwise transfer scores between 22 popular NLP tasks into TaskWeb, 2) we use TaskWeb to investigate properties such as commutativity and transitivity of pairwise transfer, 3) we propose an effective method to approximate transferability to a new target task using its examples, and 4) we demonstrate the effectiveness of our method in both single-task and multi-task settings.
>
> Below we first provide our answers to the two weakness points proposed by the reviewer.
>
> **1. About the cost of adding a new task to TaskWeb (Reasons to Reject 1)**
>
> > Although the research method is insightful, I can hardly imagine a situation where future research does not fall into the categories of TaskWeb … It is hard to first compute pair-wise transferability of any given set of tasks.
>
> Our findings in this paper demonstrate that it is possible to select helpful source tasks for new target tasks without having to compute pairwise transfer scores. More specifically, refer to the TaskShop row in Table 3 of the paper, which removes access to pairwise transfer scores for the target task. Our experiments indicate that for new target tasks that are sufficiently similar to tasks within TaskWeb, we can select helpful source tasks without prior transferability scores for the new task. One example of such a target task could be a scientific QA task which involves reasoning and question answering skills that are well addressed by TaskWeb. On the other hand, a machine translation task would be less similar to tasks in TaskWeb. That being said, the results of our experiments indicate that computing pairwise transfer scores for a new target task is not needed if the task shares sufficiently similar traits with tasks in TaskWeb.
>
> Meanwhile, we acknowledge that as language model capabilities improve over time, the scope of problems addressed by these models evolves and entirely new tasks are introduced (e.g., long-form question answering). For target tasks that considerably differ from tasks in TaskWeb, we believe it is worthwhile to compute pairwise transfer scores for new tasks that are of interest to the research community. Even if initially computing the pairwise transfer scores takes a non-trivial amount of compute, we would like to point out that the cost can be amortized over subsequent usage of the scores for different purposes. Moreover, since TaskWeb is designed to cover diverse task categories, it becomes easier to add new tasks as more tasks are added. We thank the reviewer for raising this question - we will add these explanations to the discussion and limitation sections during subsequent iterations of our draft.
>
> **2. About a suggested baseline trained on the tasks with the same category (Reasons to Reject 2)**
>
> > It would be better to have a baseline which just uses tasks within the same category as source tasks (i.e., some heuristic baselines).
>
> We thank the reviewer for the helpful suggestion. In fact, we already have results for a related experiment (which was not included in the submission draft due to space limitation) where we paired each target task with the most similar source task in terms of pairwise transferability and followed the multi-task setup presented in Section 5.2 and Table 3 of our paper. Here, 9 out of 11 target tasks use source tasks from the same task category as suggested by the reviewer, including CosmosQA to HellaSwag, CosmosQA to Story Cloze, SocialIQA to Winogrande and Winogrande to COPA for commonsense and SNLI to ANLI, ANLI to CB and ANLI to RTE for NLI/entailment. Our evaluation results indicate that this setup underperforms the top-5 multitask training sets chosen according to TaskShop or TaskWeb. For clarity, below are the results of our experiment - we will include these results into the final version of the paper if deemed necessary.
>
> | Method           | ANLI-R1 | ANLI-R2 | ANLI-R3 | CB    | COPA  | HellaSw | RTE   | Story Cloze | WiC   | Winogr. | WSC   | Mean  |
> |------------------|---------|---------|---------|-------|-------|---------|-------|-------------|-------|---------|-------|-------|
> | Top-1 Similarity | 40.83   | 34.53   | 38.08   | 79.17 | 88.17 | 42.04   | 74.56 | 93.97       | 50.38 | 52.6    | 36.54 | 57.35 |
> | Top-5 TaskShop   | 42.86   | 36.15   | 41.41   | 84.52 | 86.08 | 41.94   | 76.73 | 94.04       | 51.49 | 53.0    | 59.4  | 60.69 |
> | Top-5 TaskWeb    | 40.16   | 36.15   | 42.15   | 82.24 | 85.25 | 43.73   | 77.71 | 92.69       | 50.75 | 55.84   | 62.82 | 60.86 |
>
> - *For top-1 similarity, we randomly sample 10,000 examples from the source task to set the number of training examples equal to our other multi-task training sets.*
> - *The results for the Top-5 TaskShop and TaskWeb rows are taken from Table 3 in our paper.*
>
> Next we provide our responses to the set of questions by the reviewer, which we will clarify in the next iteration of our draft.
>
> > Q: Why does T have to be directional while F does not? It seems like both the text-embedding method and the LLM method are symmetric when acting as F?
>
> In our description of TaskShop, T refers to the pairwise transfer scores between a pair of tasks provided in TaskWeb and F refers to the off-the-shelf task selection method. To clarify our vocabulary, directional means that the pairwise transfer relies upon the ordering of the source and target tasks and therefore is asymmetric, and vice versa. Our pairwise transfer scores (T) are directional as they are based on the (source -> target) transfer - refer to our commutativity analyses in line 258 and Figure 13. On the other hand, our off-the-shelf task selection methods (F) are invariant to the (source, target) orderings and are symmetric (e.g., embedding-based method), which means that they are not directional.
>
> > Q: Line 442: for ANLI-R1/R2, we apply the subset of tasks chosen for ANLI-R3 for the upper baseline. Can you explain this in more detail? Which baseline is the "upper baseline"? Besides, further explanations may also needed for the story cloze task.
>
> Our upper baseline refers to the last row in Table 3 for the multi-task experiments, which consists of choosing the top-5 source tasks for each target task according to pairwise (source->target) transfer scores in TaskWeb. As discussed in our first response (1. Cost of adding a new task), computing pairwise transfer scores among all tasks is expensive, and in this experiment, we explore the efficacy of estimating transferability in multi-task settings without relying on pairwise transfer scores. Naturally, selecting source tasks based on pairwise transfer scores becomes our upper baseline. As ANLI-R1 and ANLI-R2 are not included in TaskWeb, we approximate the performance for choosing the source tasks using TaskWeb by using the source tasks chosen for ANLI-R3 instead of omitting evaluation results for ANLI-R1 and ANLI-R2.
>
> > Q: In the multi-task transfer setting (table 3), if it is a zero-shot setting, where do the 32 examples come from? Are they from the training set with labels removed? If so, can it be called zero-shot?
>
> We acknowledge that for our evaluation setup, we assume access to 32 labeled examples of the target task for using task selection methods between the source and target tasks. When we refer to zero-shot, we mean that we do not fine-tune the model parameters to the specific target task - we will make sure to further clarify this in subsequent refinements of the draft. Our reason for performing evaluation on the target task without any further finetuning is to remove confounding factors that may occur from finetuning the multi-task model on the target task.

---

### Meta-Review · Area_Chair_jGih · 2023-09-08

**Recommendation:** 4

**Metareview:**

The paper addresses source task selection for transfer learning. In particular, the authors construct a large-scale benchmark for pairwise task transfers for 22 NLP tasks, and introduce a method that predicts the transferability from a source task to a target task. The experimental results show the effectiveness in single-task and multi-task transfer settings.

Pros / Strengths:
- Research questions and proposed method are sound and well-designed
- Paper is well written
- Experiments are comprehensive and show the effectiveness of the proposed approach
- Observations on the commutativity and transitivity are insightful and potentially valuable to the community

Con / Weaknesses:
- Scalability: The authors acknowledged in their response that a new target task needs to share "sufficiently similar traits" with the tasks in TaskWeb for their method to work. While this is also briefly mentioned in the limitation section, the authors should elaborate more on which steps are necessary (and how computationally intense those steps might be) if a target task is not sufficiently similar to the tasks in TaskWeb
- Simple baselines missings

Action items for improved version of paper:
- Add simple heuristic baselines
- Clearly define limitations (see above)
- Clearly define "zero-shot" setup (having access to labeled examples of a target task is typically not refered to as zero-shot)
- Add statistical significance tests

---

### Decision · Program_Chairs · 2023-10-07

**Decision:**

Accept-Main

**Comment:**

The paper addresses source task selection for transfer learning. In particular, the authors construct a large-scale benchmark for pairwise task transfers for 22 NLP tasks, and introduce a method that predicts the transferability from a source task to a target task. The experimental results show the effectiveness in single-task and multi-task transfer settings.

Pros / Strengths:
- Research questions and proposed method are sound and well-designed
- Paper is well written
- Experiments are comprehensive and show the effectiveness of the proposed approach
- Observations on the commutativity and transitivity are insightful and potentially valuable to the community

Con / Weaknesses:
- Scalability: The authors acknowledged in their response that a new target task needs to share "sufficiently similar traits" with the tasks in TaskWeb for their method to work. While this is also briefly mentioned in the limitation section, the authors should elaborate more on which steps are necessary (and how computationally intense those steps might be) if a target task is not sufficiently similar to the tasks in TaskWeb
- Simple baselines missings

Action items for improved version of paper:
- Add simple heuristic baselines
- Clearly define limitations (see above)
- Clearly define "zero-shot" setup (having access to labeled examples of a target task is typically not refered to as zero-shot)
- Add statistical significance tests